# LT-Soups: Bridging Head and Tail Classes via Subsampled Model Soups

**Masih Aminbeidokhti[1,*]**    **Subhankar Roy[2]**

**Eric Granger[1]**    **Elisa Ricci[3,4]**    **Marco Pedersoli[1]**

[1]École de technologie supérieure, [2]University of Bergamo
[3]University of Trento, [4]Fondazione Bruno Kessler (FBK)

## Abstract

Real-world datasets typically exhibit long-tailed (LT) distributions, where a few head classes dominate, and many tail classes are severely underrepresented. While recent work shows that parameter-efficient fine-tuning (PEFT) methods like LoRA and AdaptFormer preserve tail-class performance on foundation models such as CLIP, we find that they do so at the cost of head-class accuracy. We identify the head-tail ratio, the proportion of head to tail classes, as a crucial but overlooked factor influencing this trade-off. Through controlled experiments on CIFAR100 with varying imbalance ratio ($\rho$) and head-tail ratio ($\eta$), we show that PEFT excels in tail-heavy scenarios but degrades in more balanced and head-heavy distributions. To overcome these limitations, we propose LT-Soups, a two-stage model soups framework designed to generalize across diverse LT regimes. In the first stage, LT-Soups averages models fine-tuned on balanced subsets to reduce head-class bias; in the second, it fine-tunes only the classifier on the full dataset to restore head-class accuracy. Experiments across six benchmark datasets show that LT-Soups achieves superior trade-offs compared to both PEFT and traditional model soups across a wide range of imbalance regimes.

## 1   Introduction

In machine learning, balanced class distributions are often assumed in both theory and practice [12, 73, 28]. However, real-world datasets frequently deviate from this assumption, exhibiting severe class imbalance where a few head classes dominate and tail classes remain significantly underrepresented [56, 19, 34]. This imbalance poses a fundamental challenge: models must learn effectively from limited tail-class data while preserving overall robustness [9].

Recent advances in vision-language foundation models, particularly CLIP [43], have introduced promising tools for addressing class imbalance. Trained on large-scale, diverse datasets, CLIP demonstrates strong robustness to distributional shifts and has become a popular backbone for long-tailed recognition [59, 58, 37, 55, 35]. Building on this, Shi et al. [50] achieve state-of-the-art results by applying parameter-efficient fine-tuning (PEFT) methods such as Low-Rank Adaptation (LoRA) [20] and AdaptFormer [8], in combination with logit adjustment (LA) loss [40, 46], which incorporates class priors by adding a class-dependent offset to the logits. While this PEFT-based approach improves overall and tail-class accuracy, they observe that it still underperforms full fine-tuning in certain regimes.

---

*Correspondence to: masih.aminbeidokhti.1@ens.etsmtl.ca. Code at https://github.com/Masseeh/LT-Soups.

39th Conference on Neural Information Processing Systems (NeurIPS 2025).

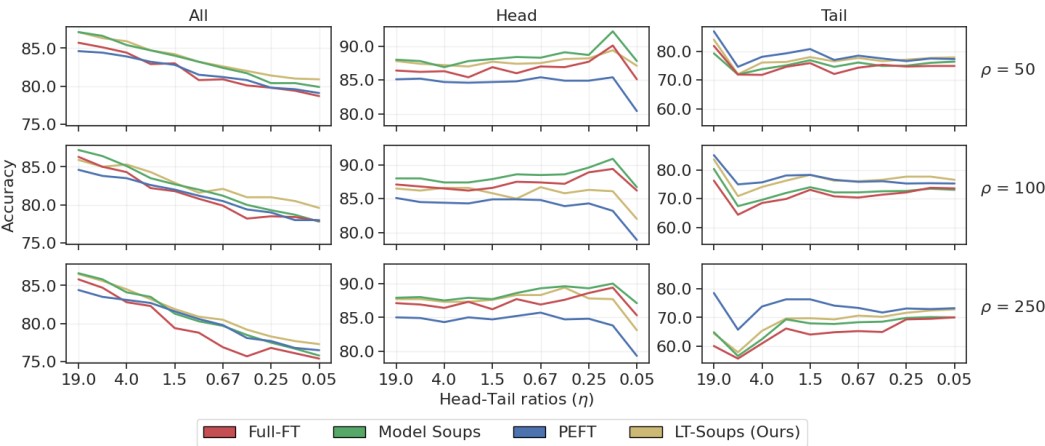

Figure 1: Performance of baselines and LT-Soups on the CIFAR100 benchmark varying $\rho$ and $\eta$. While full fine-tuning generally outperforms PEFT on head classes, PEFT demonstrates superior performance on tail classes. In contrast, our approach maintains robust accuracy across all imbalance settings, showing resilience to shifts in both the sample distribution and class structure.

These observations motivate a deeper investigation into when and why PEFT is effective. To this end, we construct a controllable long-tailed benchmark based on CIFAR100 that allows systematic variation in both the sample counts across classes, quantified by the imbalance ratio ($\rho$), and the number of classes above or below a sample threshold, defined as the head-tail ratio ($\eta$). This setup enables a more fine-grained analysis of imbalance structures. Within this framework, we compare two full fine-tuning strategies against a PEFT baseline: full fine-tuning with logit adjustment (LA) and model soups [60], which averages the weights of multiple LA-trained models initialized with different seeds and hyperparameters. Results from this benchmark confirm previous findings: on average, PEFT improves overall accuracy (80.8 vs. 81.2) and tail-class accuracy (76.4 vs. 70.2) compared to full fine-tuning, but at the cost of degraded head-class performance (87.0 vs. 84.3). A detailed breakdown in Figure 1 shows that PEFT is especially effective in tail-heavy scenarios, where rare classes dominate ($\eta \ll 1$), but its performance declines as the class structure becomes more balanced or head-heavy, highlighting its limited robustness to shifts in class structure.

This highlights a key trade-off: PEFT helps prevent overfitting and supports tail classes, but lacks adaptability in more balanced and head-heavy settings. Conversely, full fine-tuning offers stronger adaptation but requires careful regularization. Model soups offer a middle ground by averaging models trained with different seeds and hyperparameters [29, 27], but our experiments show that traditional soups, built from models trained on the same imbalanced dataset, still underperform in tail-heavy cases, as they tend to overemphasize head-class performance due to the dominance of high imbalance ratios.

To address these limitations, we introduce LT-Soups, a two-stage model soups framework designed to deliver robust performance across diverse imbalance scenarios, jointly characterized by the imbalance ratio ($\rho$) and the head-tail ratio ($\eta$). Unlike PEFT, which performs well primarily in tail-heavy settings, LT-Soups consistently achieves strong results across tail-heavy, balanced, and head-heavy class structures (Figure 1). In the *first* stage, LT-Soups constructs a weight ensemble by averaging multiple fully fine-tuned models, each trained on a subset exhibiting a distinct imbalance ratio, collectively spanning a spectrum of imbalance ratios. The aim is to create models that "specialize" for each of these imbalance ratios, when averaged, promote a balanced representation that performs well on both the head and the tail classes. To recover any head-class information lost during subsampling, the *second* stage fine-tunes only the classifier on the full dataset using class-balancing techniques. By seamlessly combining the adaptability of full-rank optimization, favouring the head-classes, and the robustness of weight ensembling for the tail-classes, LT-Soups strikes a better trade-off than PEFT and model soups, and thus bridges the head and the tail classes.

Our contributions are threefold: **(1)** We introduce a dual-axis framework for characterizing class imbalance using both the imbalance ratio ($\rho$) and head-tail ratio ($\eta$), and show how they jointly affect performance. **(2)** We propose **LT-Soups**, a novel two-stage approach that mitigates representation

bias and adapts effectively across a broad range of imbalance structures. **(3)** We conduct extensive experiments on five benchmark datasets and show that while existing LT methods perform well only under specific imbalance configurations, our approach consistently delivers robust, all-around performance across a wide range of imbalance scenarios.

## 2 Related Work

**Imbalanced Learning.** Class imbalance has traditionally been tackled through oversampling minority classes, undersampling majority classes, or applying reweighted loss functions such as focal loss or logit adjustment (LA) [7, 17, 40]. While effective in certain settings, these techniques often struggle under overparameterized models [66]. Decoupled training frameworks further refine this process by separating representation learning and classifier training [25, 67], assuming biases lie primarily in the classifier layer. However, this assumption breaks down when adapting foundation models, as full fine-tuning can lead to catastrophic forgetting and degraded generalization for both head and tail classes [41, 50].

Ensemble-based methods address class imbalance by combining experts trained on diverse data distributions [4, 53]. Examples include BBN [74] and RIDE [57], which use architectural branching or dynamic routing, and LFME [62], which employs group-wise distillation. Mixture-of-Experts approaches such as SADE [68], Mdcs [69], and DirMixE [65] merge experts trained with different logit adjustments (e.g., uniform, long-tail, inverse long-tail). Unlike these methods that require all experts at inference, LT-Soups collapses multiple fine-tuned models into a single network via weight averaging, offering an inference-efficient alternative. While prior works rely on specialized architectures and heuristic expert definitions, our approach retains architectural simplicity by using parallel fine-tuning on controlled subsamples and model averaging to preserve both the generalization and efficiency of the foundation model.

CLIP and other vision-language models exhibit inherent robustness to class imbalance, largely due to the diversity of their pretraining data [59]. This robustness has been further extended through techniques such as prompt tuning [13], retrieval-based augmentation [35], and joint vision-language training paradigms [37, 58]. While these methods improve adaptation to long-tailed distributions, Shi et al. [50] show that PEFT combined with logit adjustment (LA) loss achieves state-of-the-art performance by selectively adapting CLIP's pretrained features. However, they also observe that this comes at the cost of reduced head-class accuracy.

In this work, we demonstrate that PEFT is particularly effective in tail-heavy scenarios, but its performance diminishes as the class structure becomes more balanced or skews toward head-class dominance. To overcome this limitation, we propose a method designed to maintain robust performance across the entire long-tail distribution spectrum. Our approach merges models trained on a subset exhibiting a distinct imbalance ratio, collectively spanning a spectrum of imbalance ratios, enabling the final model to achieve balanced accuracy across both head and tail classes.

**Model Merging.** Methods based on model merging, or weight averaging, has emerged as a practical strategy for reducing communication overhead in federated and distributed settings [39, 15], improving robustness to distribution shifts [60], and enhancing generalization through techniques like SWA and EMA [23, 54]. Recent efforts also apply merging for continual learning and RLHF fine-tuning [1, 45]. Yet, to our knowledge, model merging has not been explored for imbalanced classification.

## 3 A Closer Look at Imbalanced Learning with Foundation Models

### 3.1 Preliminaries

Given training data $\mathcal{D} = \{(\boldsymbol{x}_i, y_i)\}_{i=1}^N$, where $\boldsymbol{x}_i$ denotes an input sample and $y_i \in \mathcal{C}$ is its corresponding class label from a set of $K = |\mathcal{C}|$ classes. Let $n_j$ denote the number of training samples for class $j$, and let the total number of training samples be $N = \sum_{j=1}^K n_j$. Without loss of generality, we assume that classes are sorted in decreasing order of frequency, *i.e.*, if $i < j$, then $n_i \geq n_j$. In the imbalanced setting considered here, the most frequent class is significantly larger than the rarest one, such that $n_1 \gg n_K$. To quantify this imbalance, we define the imbalance ratio as

$\rho = n_K/n_1$. Following [34], we categorize classes with more than 100 training samples ($n_j > 100$) as *head* classes, and the rest as *tail* classes.[2] Since we aim to achieve balanced performance across all classes, we report BalAcc $= \frac{1}{|\mathcal{C}|} \sum_{c \in \mathcal{C}}$ Accuracy($c$) which equally weights performance on each class.

Our model is composed of two main components: a feature extractor and a classification head. For feature extraction, we adopt the CLIP vision encoder, implemented using a Vision Transformer (ViT) [14] and parameterized by $\theta$. The representation for input $\boldsymbol{x}$ is given by $f_I(\boldsymbol{x}; \theta) = \boldsymbol{z}$ where $\boldsymbol{z}$ is the extracted feature vector. The final class prediction is computed as $\hat{y} = \arg\max g(\boldsymbol{z}; \omega)$ where $g$ denotes the prototypical classification head with parameters $\omega$ constructed from the CLIP text encoder. (see Appendix D for the full details).

Previous work suggests that training with standard *Cross-Entropy* loss with instance-balanced sampling often leads to head-class bias due to class imbalance [5, 17]. *Logit Adjustment* (LA) [40] addresses this by adding a class-dependent offset to the logits, thereby correcting for prior class frequencies as follows:

$$\ell_{LA}(y, g(\boldsymbol{z})) = -\log \frac{\exp(g_y(\boldsymbol{z}) + \log \pi_y)}{\sum_{y' \in \mathcal{C}} \exp(g_{y'}(\boldsymbol{z}) + \log \pi_{y'})} \tag{1}$$

where $g_y$ denotes the predictive logit of model on class $y$ and $\pi \in \Delta_y$ are estimates of the class priors $\mathbb{P}(y)$ based on the empirical class frequencies on the training data $D$. However, Shi et al. [50] observed that when fine-tuning CLIP models from pretrained weights using LA (referred to as *Full-FT*), the resulting class-conditional distributions can become inconsistent, particularly for tail classes. To mitigate this, they advocate for methods that preserve proximity to the pretrained initialization, leveraging PEFT strategies such as LoRA and AdaptFormer.

### 3.2 Characterizing Imbalanced Distribution with Head-Tail Ratio

In practice, class imbalance can manifest in diverse structural forms. While the imbalance ratio ($\rho$) is a standard metric for quantifying distributional skew, we show that it is insufficient to fully capture the complexity of long-tailed distributions. As a complementary measure, we introduce the head-tail ratio ($\eta$), which reflects the relative number of head versus tail classes and emphasizes the underlying class structure.

**Definition 1.** Let $\mathcal{H} = \{c \mid n_c > \tau\}$ and $\mathcal{T} = \{c \mid n_c \leq \tau\}$ denote the sets of head and tail classes, respectively, based on a sample threshold $\tau$. Let $H = |\mathcal{H}|$ and $T = |\mathcal{T}|$ be the number of head and tail classes. The head-tail ratio is then defined as $\eta = \frac{H}{T}$.

To investigate the joint effect of $\rho$ and $\eta$ on model performance, we construct a synthetic benchmark based on CIFAR100, where both parameters are systematically varied. For a fixed $\eta$, classes are partitioned into head and tail groups, and within each group, sample sizes are assigned following an exponential decay distribution. This procedure is repeated across 11 values of $\eta$, ranging from 19 (head-heavy) to 0.05 (tail-heavy), and for $\rho \in \{50, 100, 250\}$. In these configurations, head-class sample sizes range from 500 to 101, and tail-class sizes from 100 to 2 (see Figure 7a in the Appendix for visualization).

Figure 2 presents performance trends marginalized over varying $\rho$, $\eta$, and their joint effects. The results reveal that no single method consistently dominates; instead, the best-performing approach shifts depending on the imbalance configuration. In tail-heavy regimes (low $\eta$), PEFT methods excel due to their ability to retain generalizable pretrained features for underrepresented classes. Conversely, in head-heavy settings (high $\eta$), full fine-tuning becomes more advantageous, leveraging its flexibility to fit the dominant head-class structure. These trends underscore the need for methods that can adapt effectively across the full spectrum of imbalance scenarios.

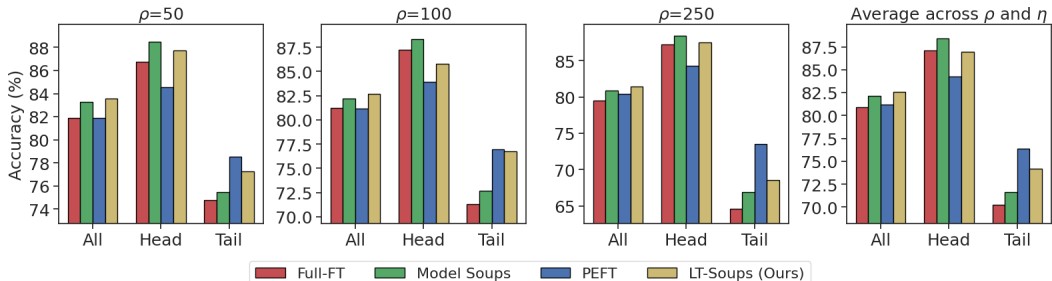

Figure 2: Marginalized performance of baselines, including LT-Soups, on CIFAR100 across varying $\rho$ and $\eta$. The first three columns average over $\eta$ for each $\rho$; the last column averages over all configurations. Refer to Figure 1 for the detailed results.

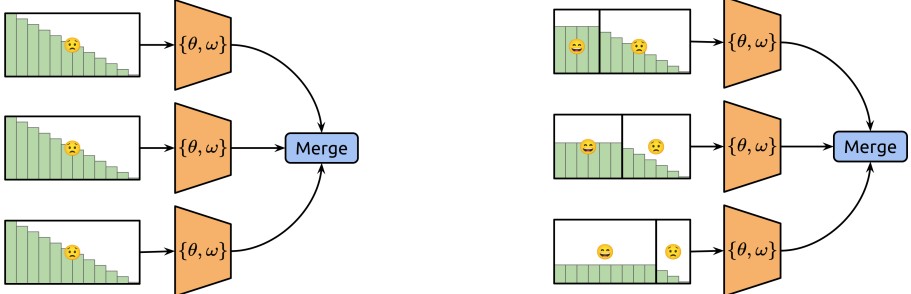

(a) Each model within Model Soups fine-tunes the full training set.

(b) Each model within LT-Soups fine-tunes a subset of less imbalanced data from the full training set.

Figure 3: Comparison between Model Soups and LT-Soups. (a) Model Soups merges models fine-tuned on full, severely imbalanced training data. (b) LT-Soups merges models fine-tuned on subsets with increasingly higher imbalance ratios to preserve pretrained features while adapting to class distribution shifts.

## 4   LT-Soups: Imbalanced Learning by Subsampled Model Averaging

The preceding toy experiment illustrates that the optimal method depends on the underlying class structure. In head-heavy distributions, full fine-tuning is particularly effective, as it adjusts all model parameters to capture the rich structure of frequent classes. In contrast, when tail classes dominate, PEFT approaches like LoRA and AdaptFormer (as used in LIFT [50]) perform better by preserving pretrained representations that generalize well under limited supervision. Motivated by this trade-off, our goal is to design a method that maintains balanced performance across both extremes, regardless of the imbalance pattern.

Ensemble methods such as model soups [60] (Figure 3a) have demonstrated effectiveness in improving both overall and minority-class performance [29, 27]. However, as shown in the previous section, traditional soups, while outperforming single-model fine-tuning, remain suboptimal for the full spectrum of long-tailed distributions. This is especially true in tail-heavy scenarios, where they tend to overemphasize head-class performance due to the dominance of high imbalance ratios in the training data. We address this limitation with LT-Soups, a soups-based framework specifically designed for long-tailed distributions. Each model in LT-Soups is fine-tuned on a subset of the training data with a distinct, reduced imbalance ratio (Figure 3b). While such subsampling enhances tail-class learning [6], it may omit valuable head-class information [26]. To balance this, we train models on subsets with gradually increasing imbalance levels, allowing each model to specialize in different regions of the long-tail spectrum. The final model is obtained by averaging these specialized models. To further recover any lost head-class information, we introduce a second stage where only the classifier head is fine-tuned on the full dataset using a class-balanced objective (e.g., LA loss).

---

[2]For simplicity, we initially group all low-resource classes into a single tail category. In the experimental section, we further subdivide the tail into *medium-shot* and *few-shot* groups for consistency with prior work.

---
**Algorithm 1** LT-Soups (Parallelizable Pseudocode)
---
1: **Input:** $\theta_0$ pre-trained weights, full training data $D$, $\{D_{\rho_n}\}_{n=1}^{MN}$ subsets with $N$ imbalance ratios $\rho_n$ and $M$ bootstrapping per $\rho_n$, $\lambda$ merging interpolation.
2: *Training*: **for all** $n = 1$ to $MN$ **in parallel do**
3: $\qquad\qquad \theta_n \leftarrow \text{FineTune}(\theta_0, D_{\rho_n})$
4: *Prepare models*: Sort models in an increasing order of $\rho_n$
5: *Weight Averaging*: $\forall n = 1$ to $N$, $\theta_n = (1 - \lambda)\theta_n + \lambda\theta_{n-1}$
6: Re-train final classifier on full $D$
---

Pseudocode for LT-Soups is provided in Algorithm 1, and the remainder of this section details the full procedure.

**Balanced representation.** Subsampling is a common strategy for addressing class imbalance by reducing overrepresented head-class samples [17, 6]. However, aggressive subsampling can discard useful head-class information, degrading overall performance [26, 7, 49]. To address this, we propose *progressive subsampling*, which incrementally increases the imbalance ratio across subsets. Each model is fine-tuned on a subset with a specific ratio, preserving tail-class data while managing head-class underutilization. We construct the subset sequence as $\left\{ D_{\rho_i} \mid \rho_i = 2^i, \ i \in \{0, 1, 2, \ldots, \lceil \log_2(\rho) \rceil\} \right\}$ where $D_{\rho_i}$ is a dataset with imbalance ratio $\rho_i$. This yields a sparse sequence of subsets with exponentially increasing imbalance ratios, ensuring broad coverage while limiting the number of models in the soups procedure. In practice, we retain only the first $N$ subsets, as extremely high imbalance ratios tend to overly favor head classes and degrade tail-class performance.

The resulting $N$ models, each trained on a different subset, are merged using a recursive interpolation strategy. Given weights $\{\theta_n\}_{n=0}^N$, where $\theta_0$ is the pretrained model, LT-Soups recursively combines models via $\theta_n = (1-\lambda)\theta_n + \lambda\theta_{n-1}$. The interpolation coefficient $\lambda$ controls knowledge retention from previous stages. This procedure ensures (1) proximity to the pretrained model $\theta_0$, preserving CLIP's zero-shot capabilities [61], and (2) smooth integration of head and tail class representations [74]. As shown in Section 5.3, LT-Soups exhibits partial insensitivity to the choice of loss function, owing to the balance introduced by subsampling and model averaging. However, since each subset remains mildly imbalanced, albeit less so than the full training set, applying LA loss during fine-tuning further mitigates the effects of label distribution shifts.

**Variance reduction.** While weight averaging and subsampling help mitigate head-class dominance, fine-tuning large pretrained models can still lead to degradation in tail-class performance [59]. To address this, we maintain an exponential moving average (EMA) of model weights via $\theta_{ema} = (1 - \mu) \cdot \theta_{ema} + \mu \cdot \theta$, with a momentum coefficient $\mu = 0.99$. EMA acts as a regularizer during training [21], promoting convergence to flatter minima [23], which has been shown to enhance generalization, particularly for underrepresented classes.

Since subsampling reduces data per subset and introduces variance, we adopt a bootstrapping strategy inspired by bagging [2]: for each subset, we train $M$ models on different bootstrap samples and uniformly average their weights. This stabilizes learning and yields more robust representations.

**Classifier re-training.** To further recover head-class information lost during subsampling, we perform a final fine-tuning stage on the classifier head using the full training set. The backbone is frozen to preserve merged representations, and LA loss is applied to adjust decision boundaries based on label frequencies. This step improves head-class accuracy without harming tail-class performance, similar to calibration in two-stage LT methods [25].

## 5 Experiments

### 5.1 Datasets and evaluation protocol.

We evaluate our method on both synthetically constructed and naturally occurring long-tailed (LT) datasets. For synthetic benchmarks, we use CIFAR-100-LT, ImageNet-LT, and Places-LT—long-tailed variants derived from their balanced counterparts by sampling class instances according

to Pareto or exponential decay distributions [34]. These datasets exhibit sample counts ranging from 1,280 to as few as 5 images per class. For real-world evaluation, we include iNaturalist 2018 (8,142 classes, 437.5K images) and NIH-CXR-LT (20 classes, 88.5K images), which reflect different imbalance structures, with approximately 10% and 90% head classes, respectively. To assess performance across the long-tail spectrum, we also report the average accuracy across all five datasets. Following [34], we evaluate separately on many-shot (>100 samples), medium-shot (20–100), and few-shot (<20) class subsets. For ablation analysis, we use TinyImageNet-LT, which contains 200 classes with sample counts ranging from 500 in head classes to 5 in tail classes. To conserve space, we present only CLIP-based results in the main text; additional implementation details and extended results are included in Appendix D.

Table 1: Comparison with state-of-the-art methods on synthetic LT distributions.

| Methods | CIFAR100-LT $\rho$=100 $\eta$=0.54 | | | | Places-LT $\rho$=996 $\eta$=0.55 | | | | ImageNet-LT $\rho$=256 $\eta$=0.62 | | | |
| | All | Many | Med. | Few | All | Many | Med. | Few | All | Many | Med. | Few |
|---|---|---|---|---|---|---|---|---|---|---|---|---|
| BALLAD [37] | - | - | - | - | 49.5 | 49.3 | 50.2 | 48.4 | 75.7 | 79.1 | 74.5 | 69.8 |
| Decoder [58] | - | - | - | - | 46.8 | - | - | - | 73.2 | - | - | - |
| LPT [13] | - | - | - | - | 50.1 | 49.3 | 52.3 | 46.9 | - | - | - | - |
| Linear Probing | 70.0 | 77.2 | 71.1 | 60.4 | 48.8 | 48.8 | 49.7 | 47.1 | 74.2 | 77.8 | 73.3 | 67.4 |
| Full-FT | 79.6 | 88.1 | 79.9 | 69.3 | 46.6 | 49.9 | 46.3 | 41.4 | 73.9 | 79.8 | 71.9 | 63.9 |
| cRT [25] | 78.8 | 89.7 | 79.7 | 65.1 | 44.4 | 51.0 | 43.1 | 35.4 | 72.6 | 81.1 | 70.6 | 56.1 |
| PEFT [50] | 81.3 | 85.2 | 80.9 | 77.1 | 51.5 | 51.3 | 52.2 | **50.5** | 77.0 | 80.2 | **76.1** | **71.5** |
| Model Soups [60] | 82.1 | **89.9** | 82.2 | 73.0 | 49.4 | **51.7** | 50.0 | 43.7 | 76.0 | **81.5** | 74.5 | 65.5 |
| LT-Soups (Ours) | **83.5** | 88.2 | **83.5** | **78.0** | **51.7** | 51.2 | **52.8** | 50.3 | **77.4** | 81.2 | **76.1** | 70.7 |

Table 2: Comparison with state-of-the-art methods on real-world LT distributions.

| Methods | NIH-CXR-LT $\rho$=6491 $\eta$=5.66 | | | | iNaturalist 2018 $\rho$=500 $\eta$=0.11 | | | |
| | All | Many | Med. | Few | All | Many | Med. | Few |
|---|---|---|---|---|---|---|---|---|
| BALLAD [37] | 34.5 | 36.7 | 38.9 | 20.8 | 49.5 | 49.3 | 50.2 | 48.4 |
| Decoder [58] | - | - | - | - | 59.2 | - | - | - |
| LPT [12] | - | - | - | - | 76.1 | - | - | 79.3 |
| Linear Probing | 17.5 | 13.3 | 21.1 | 16.7 | 60.4 | 48.9 | 60.0 | 63.9 |
| Full-FT | 38.0 | 43.8 | **41.5** | 20.0 | 76.1 | 75.7 | 76.9 | 75.3 |
| cRT [25] | 37.7 | 42.9 | 39.3 | 25.0 | 44.4 | 51.0 | 43.1 | 35.4 |
| PEFT [50] | 38.5 | 43.3 | 40.4 | 25.5 | **79.1** | 72.4 | **79.0** | **81.1** |
| Model Soups [60] | 38.0 | **45.6** | 40.2 | 20.0 | 76.4 | **77.1** | 76.8 | 75.6 |
| LT-Soups (Ours) | **39.3** | 42.4 | 40.7 | **30.9** | 78.2 | 76.7 | 78.5 | 78.2 |

## 5.2 Main results

**Synthetic LT datasets.** Table 1 presents the accuracy of LT-Soups on three benchmark datasets with synthetically induced long-tail distributions: CIFAR100-LT, Places-LT, and ImageNet-LT. Our method outperforms all state-of-the-art baselines in overall accuracy on every dataset. Notably, LT-Soups surpasses Model Soups and PEFT, the most competitive baselines. While PEFT achieves competitive performance on tail classes through low-rank adaptation, it often does so at the cost of many-shot accuracy, especially in CIFAR100-LT, where LT-Soups maintains strong tail accuracy (78.0) without sacrificing performance on many-shot (88.2) or medium-shot (83.5) categories. Model Soups, on the other hand, tends to overfit many-shot categories (e.g., 89.9 on CIFAR100-LT) but underperforms on few-shot classes due to averaging independently fine-tuned models without accounting for class imbalance.

**Real-world LT datasets.** In Table 2, we evaluate LT-Soups on two naturally imbalanced datasets—NIH-CXR-LT and iNaturalist 2018—which present distinct challenges. (1) NIH-CXR-LT consists primarily of head-class (many-shot) samples but diverges significantly from CLIP's pretraining domain, as it comprises medical X-ray images. (2) iNaturalist 2018 is heavily skewed toward medium- and few-shot categories and is more closely aligned with CLIP's natural image priors. On NIH-CXR-LT, LT-Soups achieves the highest overall accuracy (39.3%), outperforming PEFT (38.5%) and delivering substantial gains in the few-shot regime (+5 points over PEFT and +10 over

Model Soups). On iNaturalist, where PEFT performs strongly (79.1% overall), LT-Soups remains competitive (78.2%) while offering more balanced accuracy across many-, medium-, and few-shot subsets.

**Full spectrum results.** The datasets in our benchmark exhibit diverse long-tail characteristics, with imbalance ratios ranging from 100 to 6,491 and head-tail ratios spanning 0.11 to 5.66. To abstract away the effects of individual dataset characteristics, Figure 4 reports the average accuracy of Full-FT, Model Soups, PEFT, and LT-Soups across all five benchmarks. While PEFT performs well on medium-shot and few-shot splits, and Model Soups excels on many-shot classes, LT-Soups consistently achieves strong performance across all splits, demonstrating its robustness across the long-tail spectrum.

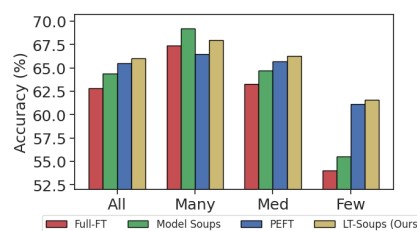

Figure 4: Average performance across 5 LT benchmarks.

## 5.3 Empirical analysis of LT-Soups

In this section, we provide a comprehensive analysis of LT-Soups from multiple perspectives (due to space constraints, some of the analysis is provided in the Appendix A).

**Effect of subsampling.** LT-Soups averages the weights of $NM$ models, where $N$ is the number of subsets used during each fine-tuning and $M$ is the number of bootstraps per subset. For TinyImageNet-LT (imbalance ratio 100 and head-tail ratio 0.3), we use $N$=8 and $M$=2. To show the impact of our proposed weight averaging scheme, we compare this with Soups-$\rho_n$ baselines that follow the same two-stage framework as LT-Soups, except in the first stage, they average 16 models, all trained on subsets with the same imbalance ratio $\rho_n$. Notably, Soups-100 aligns with the traditional model soup approach [60], where the weights of 16 fully fine-tuned models on the entire dataset are averaged. As shown in Table 3, all of the soups baselines consistently outperform full fine-tuning, regardless of the subset choice. However, results show that different imbalance ratios yield varying outcomes across head and tail categories. For example, Soups-8 achieves the highest tail accuracy of 75.0, whereas Soups-100 reaches the highest head accuracy of 85.9. Rather than optimizing for a single imbalance ratio, LT-Soups applies weight averaging across the full spectrum, effectively merging the advantages of both approaches to achieve a more balanced overall trade-off. (See Table 10 in the Appendix for a similar analysis on PEFT.)

Table 3: Comparison of LT-Soups and Soups-$\rho$ each with a total of 16 models across All, Head, and Tail accuracy.

|      | Full-FT | PEFT | Soups-1 | Soups-2 | Soups-4 | Soups-8 | Soups-16 | Soups-32 | Soups-64 | Soups-100 | LT-Soups |
|------|---------|------|---------|---------|---------|---------|----------|----------|----------|-----------|----------|
| All  | 73.2    | 77.1 | 71.7    | 75.9    | 76.0    | 77.2    | 77.2     | 77.3     | 77.9     | 77.6      | 78.6     |
| Head | 83.4    | 83.0 | 74.6    | 78.6    | 78.7    | 81.0    | 82.8     | 84.7     | 85.5     | 85.9      | 85.0     |
| Tail | 67.7    | 73.9 | 70.1    | 74.4    | 74.6    | 75.0    | 74.1     | 73.3     | 73.7     | 73.0      | 75.2     |

**Effect of classifier re-training (CR).** We found that additional final-layer tuning with logit adjustment on PEFT and Model Soups has little to no effect. Table 4 summarizes the results on TinyImageNet-LT. We hypothesize that, unlike these baselines, LT-Soups does not fully exploit the entire training set, due to the downweighting effect introduced by weight averaging. Consequently, fine-tuning the final layer helps LT-Soups recover head-class sharpness and improves overall performance.

**Component analysis.** LT-Soups is designed to balance effective task adaptation with minimal deviation from pretrained weights. Figure 5 shows the cumulative effect of its components on accuracy and weight change. Starting from Full Fine-Tuning, which causes the largest deviation from the CLIP zero-shot model (35.4), each component incrementally improves performance while reducing or controlling weight deviation. EMA offers a modest accuracy boost with minimal impact on weight shift. Subsampling and model merging significantly improve tail accuracy (+6.3) and reduce weight change to 12.7, highlighting the benefit of balanced training. Bootstrapping stabilizes training further, slightly improving head accuracy. Classifier re-training refines decision boundaries,

Table 4: Comparison of baseline methods including LT-Soups with and without classifier re-training (CR).

| Method | All | Head | Tail |
|---|---|---|---|
| PEFT | 77.1 | 83.0 | 73.9 |
| PEFT + CR | 77.0 | 83.0 | 73.8 |
| Model Soups | 77.6 | 85.9 | 73.0 |
| Model Soups + CR | 77.6 | 85.5 | 73.4 |
| LT-Soups Stage 1 | 78.1 | 84.9 | 74.5 |
| LT-Soups | **78.6** | **85.0** | **75.2** |

Table 5: Comparison of our merging strategy with uniform merging across two datasets exhibiting distinct long-tailed distributions.

| | TinyImageNet-LT | | iNaturalist 2018 | |
| | Ours | Unifrom | Ours | Unifrom |
|---|---|---|---|---|
| All | 78.6 | 78.5 | 78.2 | 74.7 |
| Many | 85.0 | 83.4 | 76.7 | 67.4 |
| Med. | 78.3 | 78.4 | 78.5 | 75.8 |
| Few | 71.5 | 72.9 | 78.2 | 75.3 |

|  | Tail / All / Head | ‖ΔW‖ |
|---|---|---|
| Full-FT | 67.7 / 73.2 / 83.4 | 35.4 |
| +EMA | 68.8 / 74.1 / 84.0 | 35.1 |
| +Sub. | 74.4 / 78.0 / 84.6 | 12.7 |
| +Bootst. | 74.5 / 78.1 / 84.9 | 11.9 |
| +Classifer. | 75.2 / 78.6 / 85.0 | 12.1 |
| PEFT | 73.9 / 77.1 / 83.0 | 10.3 |

Figure 5: Performance and weight change comparison across different stages of LT-Soups.

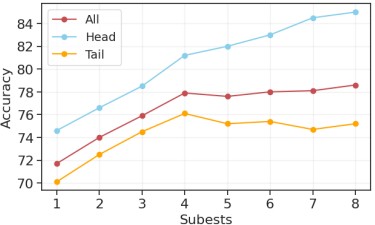

Figure 6: Performance across number of subsets, $N$, each with increasing imbalance ratios on TinyImageNet-LT.

yielding the highest overall and head accuracy. Compared to PEFT, LT-Soups shows a slightly higher weight change (12.1 vs. 10.3) but delivers better accuracy across all class groups. This reflects its ability to adapt meaningfully while preserving pretrained knowledge.

**Number of subsets.** Figure 6 illustrates the impact of the number of subsets $N$, each with increasing imbalance ratios, used in LT-Soups during fine-tuning. In this experiment, the interpolation weight $\lambda$ and $M$ are fixed at 0.7 and 2, respectively. As $N$ increases, head-class accuracy steadily improves—from 74.6 at $N$=1 to 85.0 at $N$=8—while tail-class accuracy peaks at 76.1 when $N$=3. The best overall trade-off is observed at $N$=8, indicating it as the most balanced configuration.

**Merging strategies.** LT-Soups recursively merges models trained on subsets with progressively higher imbalance ratios. One intuitive way to think about this merging procedure is to interpret it as an exponential moving average (EMA) over fine-tuned models sorted by increasing imbalance severity, with a tunable parameter that adjusts the influence of more balanced (but smaller) versus less balanced (but larger) subsets. In this section, we compare this strategy against uniform WA, which applies a simple arithmetic mean, giving equal weight to all models regardless of their imbalance level.

Table 5 confirms our hypotheses. In particular, we compare recursive WA and uniform WA across two datasets with different similarities compared to CLIP-pretrained weights (according to the zero-shot performance). On TinyImageNet-LT, which is already well-aligned with CLIP-pretrained features, there is little to no difference between the two averaging schemes. However, for datasets that require significant adaptation, such as iNaturalist2018, recursive WA yields clear benefits by leveraging information from more data-rich subsets.

Following this intuition, in all of our experiments in the paper, we use only two values for $\lambda$: 0.3 and 0.7, corresponding to high and low adaptation needs, respectively. Intuitively, when the target dataset is close to the pre-training weights, the value of the $\lambda$ becomes less important as even small datasets are enough for adaptations. However, when the shift becomes larger, subsets with more data (albeit biased towards head classes) become crucial.

**Effect of class-balance strategies.** By default, LT-Soups exhibits partial insensitivity to the choice of loss function, due to the balance introduced through subsampling and model averaging. However, in the first stage, each subset remains mildly imbalanced, though significantly less so than the full training set. Table 6 reports the impact of different class-balancing strategies used during this

Table 6: Comparison of PEFT and LT-Soups under different loss functions (CE, CB, LA).

| Methods | CE | | | CB | | | LA | | |
|---------|-----|------|------|-----|------|------|-----|------|------|
| | All | Head | Tail | All | Head | Tail | All | Head | Tail |
| PEFT | 72.6 | 85.1 | 65.9 | 75.3 | 81.0 | 72.2 | 77.1 | 83.0 | 73.9 |
| LT-Soups | 76.3 | 84.5 | 71.9 | 78.2 | 84.5 | 78.4 | 78.6 | 85.0 | 75.2 |

stage, including logit adjustment loss (LA), cross-entropy loss (CE), and class-balanced sampling (CB) [25]. Unlike PEFT, which heavily depends on LA loss for optimal performance, LT-Soups is only moderately affected by the choice of class-balancing strategies, owing to the structural regularization introduced by training on multiple, complementary subsets.

**Computational analysis.** LT-Soups involves a total of $NM + 1$ training runs: $M$ models are trained at each of the $N$ subsampling levels in the first stage, followed by a single classifier trained on the full dataset in the second stage. Two key factors mitigate the computational burden of this procedure. First, each Stage 1 model is trained on a heavily subsampled dataset, significantly smaller than the full training set. For instance, under a maximum imbalance ratio of 64, each model sees only about 65% of the full dataset (see Table 11 in the Appendix for precise values), which leads to substantially faster training times compared to full-data training. Second, all models in the first stage are trained independently, enabling full parallelization. As a result, the overall wall-clock time is bounded by the longest individual training job, typically the one using the most imbalanced subset (e.g., imbalance ratio 64). This parallel-friendly design allows LT-Soups to scale efficiently and deliver competitive performance with minimal overhead. Appendix B provides a full breakdown of computational cost.

# 6 Limitations and Future Work

While we define the head-tail ratio using a fixed sample threshold, this binary split may oversimplify the class distribution structure. A more nuanced approach could leverage the generalized Pareto distribution [47] to model imbalance with controllable location, scale, and shape parameters. We leave exploration of such parameterized formulations for future work.

# 7 Conclusion

This work introduces LT-Soups, a novel two-stage model merging framework tailored for long-tailed distributions. We identify the head-tail ratio ($\eta$) as a critical yet underexplored factor influencing model performance alongside the commonly studied imbalance ratio ($\rho$). Through comprehensive experiments, we demonstrate that existing approaches, particularly PEFT and traditional model soups, fail to generalize across the full spectrum of imbalance scenarios. In contrast, LT-Soups builds a weight ensemble by averaging fully fine-tuned models trained on subsets with varying imbalance ratios, enabling specialization across the imbalance spectrum while preserving robust representations. Extensive experiments on five benchmarks and ablations on Tiny-ImageNet-LT confirm its consistent performance across long-tailed scenarios.

**Acknowledgements.** This work was supported by Distech Controls Inc., the Natural Sciences and Engineering Research Council of Canada, the Digital Research Alliance of Canada, and MITACS. This work was also supported in part by the project SERICS (PE00000014) under the NRRP MUR program funded by the EU - NGEU.

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

## Broader Impact

Our work advances the field of long-tailed recognition by improving model performance across imbalanced datasets, which are prevalent in real-world applications such as medical imaging, wildlife monitoring, and autonomous driving. By enhancing accuracy for both head and tail classes, our method promotes fairness and inclusivity in AI systems, reducing biases toward dominant categories.


# A  Additional ablations

**Model calibration analysis.**   An inherent advantage of model merging methods is their ability to improve prediction calibration metrics. We evaluate LT-Soups against PEFT and Full-FT by measuring Negative Log-Likelihood (NLL), Expected Calibration Error (ECE) [42], and Brier score [3]. For NLL and Brier scores, we also provide category-wise results. All metrics are computed after temperature tuning on the validation set. As shown in 7, LT-Soups consistently outperforms the other methods on TinyImageNet-LT in terms of calibration.

Table 7: Calibration metrics on TinyImageNet for Full-FT, PEFT, and our LT-Soup.

| Method | Metric | Overall | Head | Tail |
|---|---|---|---|---|
| Full-FT | ECE | 1.97 | - | - |
| | Brier Score | 0.36 | 0.21 | 0.40 |
| | NLL | 1.03 | 0.63 | 1.25 |
| PEFT | ECE | 1.95 | - | - |
| | Brier Score | 0.32 | 0.23 | 0.35 |
| | NLL | 0.89 | 0.68 | 0.99 |
| LT-Soups | ECE | **1.36** | - | - |
| | Brier Score | **0.30** | **0.20** | **0.33** |
| | NLL | **0.83** | **0.59** | **0.97** |

Table 8: Performance across different values of $\lambda$ with a fixed $M=2$, on TinyImageNet-LT.

| | $\lambda=0.3$ | $\lambda=0.7$ |
|---|---|---|
| Acc | 78.3 | 78.6 |
| Head | 84.6 | 85.0 |
| Tail | 75.0 | 75.2 |

Table 9: Performance across different values of $M$ with a fixed $\lambda=0.7$, on TinyImageNet-LT.

| | $M=1$ | $M=2$ | $M=12$ |
|---|---|---|---|
| Acc | 78.2 | 78.6 | 78.8 |
| Head | 84.8 | 85.0 | 85.5 |
| Tail | 74.6 | 75.2 | 75.2 |

**PEFT compatibility.**   A natural question is whether PEFT methods can replace the full fine-tuning process in LT-Soups. To investigate this, we use LoRA as a representative approach. In the first stage, we freeze the CLIP pre-trained weights and tune LoRA parameters using the same subsets as LT-Soups. The LoRA parameters are combined with the pre-trained weights before applying our merging schema. Finally, we retrain the classifier using the LA loss. The performance on TinyImagenet-LT dataset (77.1 vs 77.2) matches that of end-to-end LoRA training. We hypothesize this outcome is due to a phenomenon observed in LLM literature [51], where LoRA introduces high-ranking singular vectors (intruder dimensions) that are absent in full fine-tuning. While these models achieve comparable task performance, they adapt less robustly to sequential tasks and diverge from the pre-training distribution.

Table 10: Effect of subsampling and classifier re-training in conjunction with the PEFT method. Each column reports PEFT fine-tuning performance on a given subsample ratio $\rho$ after classifier re-training on TinyImageNet-LT.

| $\rho$ | 1 | 2 | 4 | 8 | 16 | 32 | 64 | 100 |
|---|---|---|---|---|---|---|---|---|
| All | 73.9 | 74.3 | 74.8 | 76.2 | 76.4 | 77.1 | 77.0 | 77.0 |
| Head | 75.9 | 75.8 | 77.0 | 78.2 | 80.4 | 81.8 | 81.9 | 83.0 |
| Tail | 72.8 | 73.5 | 73.6 | 75.1 | 74.2 | 74.5 | 74.4 | 73.8 |

**Hyperparameter sensitivity.**   In addition to the number of subsets used during LT-Soups fine-tuning, two other hyperparameters impact performance: (1) $M$: the number of models trained per subset $D_{\rho_i}$, with each model bootstrapped from the same imbalance ratio $\rho_s$. (2) $\lambda$: the interpolation coefficient used during recursive weight averaging.

Table 9 shows that increasing $M$ on TinyImageNet-LT improves overall, head-, and tail-class accuracy, highlighting the benefits of ensembling across bootstrapped models. To ensure computational feasibility across all five datasets, we fix $M = 2$ in the main experiments.

Table 8 compares performance for $\lambda = 0.3$ and $\lambda = 0.7$. We observe that datasets closely aligned with CLIP's pretraining domain benefit from a larger $\lambda$, which retains more pretrained knowledge. In contrast, datasets with significant domain shifts—such as NIH-CXR-LT—perform better with smaller $\lambda$, allowing greater adaptation during model merging.

Table 11: Number of subsampling rounds ($N$), size of the largest subset relative to the full training set and $\lambda$ used for each dataset.

| Dataset | $N$ | Relative size of largest subset | $\lambda$ |
|---------|-----|--------------------------------|-----------|
| CIFAR100-LT | 5 | 67 | 0.7 |
| Places-LT | 5 | 63 | 0.7 |
| ImageNet-LT | 7 | 79 | 0.7 |
| iNaturalist | 8 | 90 | 0.3 |
| NIH-CXR-LT | 8 | 24 | 0.3 |

# B Full Computational analysis

Table 12: Comparison of methods across ImageNet-LT and CXR-LT in terms of training time, iterations, model size, memory, and accuracy.

| Method | Wall-clock Time (H-M-S) | Training Iterations | Parameters (M) | Memory (G) |
|--------|------------------------|--------------------|--------------|-----------|
| ImageNet-LT | | | | |
| Full-FT | 1:37:56 | 9060 | 87.0 | 14.5 |
| Model Soups | 1:37:56 | 9060 | 87.0 | 14.5 |
| LoRA ($rank = 64$) | 1:25:33 | 9060 | 9.0 | 13.3 |
| LT-Soups Stage 1 | 1:15:38 | 8050 | 87.0 | 14.5 |
| LT-Soups Stage 2 | 0:30:00 | 9060 | 0.7 | 2.6 |
| Full LT-Soups | 1:48:38 | – | – | – |
| NIH-CXR-LT | | | | |
| Full-FT | 0:53:43 | 5320 | 87.0 | 14.5 |
| Model Soups | 0:53:43 | 5320 | 87.0 | 14.5 |
| LoRA ($rank = 64$) | 2:14:32 | 13300 | 9.0 | 13.3 |
| LT-Soups Stage 1 | 0:12:51 | 1300 | 87.0 | 14.5 |
| LT-Soups Stage 2 | 0:19:26 | 5320 | 0.7 | 2.6 |
| Full LT-Soups | 0:32:17 | – | – | – |

Table 12 compares the computational costs of Full Fine-Tuning (Full-FT), LIFT (which employs a LoRA adapter with rank 64 applied to all MLP layers), Model Soups, and LT-Soups on the ImageNet-LT and NIH-CXR-LT datasets. All models were trained to convergence using a batch size of 128 and mixed-precision training with NVIDIA RTX 3090 GPUs (24GB VRAM), using Python 3.9.15, PyTorch 2.4.0, and CUDA 11.8. For LT-Soups, we break down the computational cost into two stages. Stage 1 involves training models independently and in parallel on subsets with different imbalance ratios. Since the subset with the highest imbalance ratio contains the most training samples, it dominates the overall wall-clock time. Stage 2 retrains only the linear classifier on the full dataset—a highly efficient step, as it updates just a single linear layer. In our experiments, we used the same number of epochs for both stages of LT-Soups.

The computational overhead of LT-Soups compared to existing methods depends heavily on dataset characteristics, particularly the original imbalance ratio. For example, in ImageNet-LT, which has an imbalance ratio of 256, the largest subset used in Stage 1 accounts for 89% of the full training data, resulting in relatively higher wall-clock time. In contrast, on NIH-CXR-LT, with a much more extreme imbalance ratio of 6401, the largest Stage 1 subset represents only 24% of the dataset, leading to a 4.4× reduction in training time compared to Full-FT (Table 11). Additionally, while full-rank

methods like Full-FT and LT-Soups typically converge within 10 epochs on CXR-LT, LIFT required 50 epochs—substantially increasing its wall-clock time despite its parameter-efficient design.

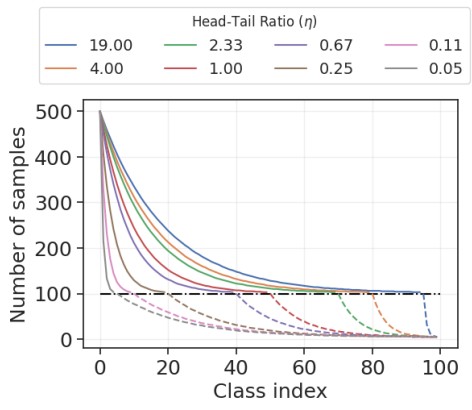
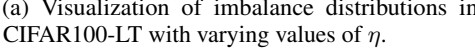
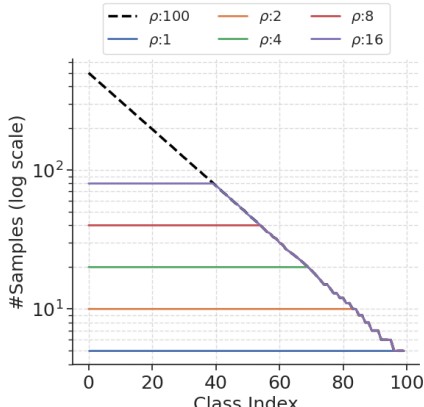

(a) Visualization of imbalance distributions in CIFAR100-LT with varying values of $\eta$.

(b) Example of subsampled distributions used in LT-Soups, with the x-axis shown on a logarithmic scale.

## C  Extended Related Work

### C.1  Imbalanced Classification

We can roughly divide progress on imbalanced classification into three groups.

**Re-sampling/Re-weighting.**    Class imbalance mitigation strategies broadly involve oversampling minority classes [7], subsampling majority classes [33], or reweighting losses [17]. Generative approaches such as DiffuLT [48] train a diffusion model on a long-tailed distribution and then uses it to generate a balanced proxy dataset for training the final classifier. Subsampling risks losing majority-class discriminative patterns, oversampling may overfit minority classes [74], and reweighting struggles in overparameterized networks [66]. Recent advances like *balanced softmax* [46] and its generalization, *logit adjustment* loss (LA) [40] address these issues by enforcing larger margins for tail classes, bridging data imbalance with geometric regularization.

**Decoupled learning.**    Decoupled learning frameworks address class imbalance through sequential training phases: representation learning via instance-balanced sampling followed by classifier refinement using class-balanced strategies [25, 67]. This paradigm assumes model biases primarily reside in the classifier layer, positing that head-tail performance gaps can be resolved through post-hoc classifier calibration [24, 64]. However, [50] show empirically that this assumption becomes invalid when fine-tuning foundation models, neglecting a tailored strategy for representation learning, degrades both head and tail class performance due to catastrophic forgetting of pre-trained features [41].

**Ensemble learning.**    Ensemble methods address data imbalance by combining specialized experts trained on complementary distributions [4, 31]. Notable approaches include: BBN's dual-branch architecture balancing original and re-sampled distributions [74]; RIDE's dynamic routing of instances to distribution-aware experts [57]; and LFME's multi-teacher distillation across many/medium/few-shot groups [62]; Reflective Learning [70] promotes consistency across training iterations by minimizing KL divergence between predictions and soft labels induced from feature similarity. While effective, these methods rely on heuristic expert specialization rules and often result in cumbersome architectures that hinder adaptation to foundation models, increase training complexity, and limit inference speed. Our work circumvents these limitations through two key innovations: (1) replacing specialized expert design with parallel fine-tuning of foundation models on controlled subsamples, and (2) employing model averaging and EMA instead of complex aggregation mechanisms. This pre-

serves the ensemble's variance-reduction benefits while maintaining the original foundation model's architectural simplicity and computational efficiency.

## C.2 Model Merging

Model merging, also sometimes referred to to weight averaging, has gained significant attention in recent years as a promising research direction [32], focusing on reducing communication costs in federated learning [39] and distributed training [15], enabling the efficient combination of multiple models without additional training [22], and enhancing model robustness in out-of-distribution scenarios [60, 44]. Early approaches like Exponential Moving Average (EMA) [54] and Stochastic Weight Averaging (SWA) [23] have been widely adopted to accelerate training convergence, stability, and enhance the generalization capabilities of deep neural networks. Recent work extends merging to sequential adaptation: Alexandrov et al. [1] mitigates catastrophic forgetting in continual pretraining via iterative merging, while Ramé et al. [45] align LLMs through multi-stage averaging during RLHF. To our knowledge, no prior work applies model merging to imbalanced recognition. Unlike existing sequential merging approaches, our framework trains multiple models in parallel on complementary subsampled distributions, a critical design choice for handling long-tailed data. We propose the first schema specifically tailored for imbalance, integrating subsampling (to retain tail-class discriminability) and bootstrapping (to stabilize head-class representations). This parallelized merging strategy directly addresses feature-space asymmetry in long-tailed distributions while maintaining computational efficiency, enabling foundation models to adapt to extreme imbalance without sacrificing pre-trained generalization.

# D    Baselines and implementation details.

We use CLIP with the ViT-B/16 backbone. Following [43], we adopt a prototypical classification head for $g$, where both features and classifier weights are $l_2$-normalized, and a temperature is applied to the logits. The parameters $\omega$ are initialized by generating text. We use descriptive prompts such as "a photo of a cat" or "a photo of a dog" to represent each class [43]. for the classes and extracting corresponding textual features using the CLIP text encoder.

We optimize the model using the AdamW optimizer [36]. The batch size is set to 128, with learning rates of $3e - 4$ for both the representation and the classification stage. A cosine decay learning rate scheduler is employed, gradually reducing the learning rate to $0.1 \cdot max\_lr$ after a warmup period spanning $max(100, 0.01 \cdot total\_steps)$ steps. The validation set of each dataset is used to select the best checkpoint. Table 11 shows the hyperparameters we used for each dataset. We select $N$ and $\lambda$ based on the validation set of each dataset and fix $M$=2 across all experiments. We report all baseline results without test-time augmentation, which offers orthogonal gains.

Table 13: Dataset details used in our work.

| Dataset | Classes | Total samples | Max samples | Min samples | $\rho$ | $\eta$ |
|---|---|---|---|---|---|---|
| CIFAR100-LT [5] | 100 | 10.8k | 500 | 5 | 100 | 0.54 |
| TinyImageNet-LT [30] | 200 | 21.5k | 500 | 5 | 100 | 0.53 |
| Places-LT [34] | 365 | 62.5k | 4980 | 5 | 996 | 0.55 |
| ImageNet-LT [34] | 1000 | 115.8k | 1280 | 5 | 256 | 0.62 |
| iNaturalist [56] | 8,142 | 437.5k | 1000 | 2 | 500 | 0.11 |
| NIH-CXR-LT [19] | 20 | 88,637 | 53260 | 12 | 6491 | 5.66 |

## D.1    Full results

Table 14: Comparison of methods for training on CIFAR100-LT.

| Methods | Backbone | Overall | Many | Medium | Few |
|---|---|---|---|---|---|
| Training from scratch | | | | | |
| LDAM [5] | ResNet-32 | 42.0 | - | - | - |
| BBN [74] | ResNet-32 | 42.6 | - | - | - |
| DiVE [18] | ResNet-32 | 45.4 | - | - | - |
| MiSLAS [72] | ResNet-32 | 47.0 | - | - | - |
| BS [46] | ResNet-32 | 50.8 | - | - | - |
| PaCo [10] | ResNet-32 | 52.0 | - | - | - |
| BCL [75] | ResNet-32 | 51.9 | - | - | - |
| Fine-tuning CLIP | | | | | |
| Linear Prob (LA) | ViT-B/16 | 70.0 | 77.2 | 71.1 | 60.4 |
| Full-FT (LA) | ViT-B/16 | 79.6 | 88.1 | 79.9 | 69.3 |
| cRT [25] | ViT-B/16 | 78.8 | 89.7 | 79.7 | 65.1 |
| PEFT [50] | ViT-B/16 | 81.3 | 85.2 | 80.9 | 77.1 |
| Model Soups [60] | ViT-B/16 | 82.1 | **89.9** | 82.2 | 73.0 |
| LT-Soups (Ours) | ViT-B/16 | **83.5** | 88.2 | **83.5** | **78.0** |

Table 15: Comparison of methods for training on Places-LT.

| Methods | Backbone | Overall | Many | Medium | Few |
|---|---|---|---|---|---|
| Training from ImageNet-1K pre-trained backbone | | | | | |
| OLTR [34] | ResNet-152 | 35.9 | 44.7 | 37.0 | 25.3 |
| cRT [25] | ResNet-152 | 36.7 | 42.0 | 37.6 | 26.4 |
| LWS [25] | ResNet-152 | 37.6 | 40.6 | 39.1 | 28.6 |
| MiSLAS [72] | ResNet-152 | 40.4 | 39.6 | 43.3 | 36.1 |
| DisAlign [67] | ResNet-152 | 39.3 | 40.4 | 39.4 | 32.9 |
| ALA [71] | ResNet-152 | 41.2 | 36.1 | 47.9 | 35.3 |
| PaCo [10] | ResNet-152 | 40.5 | 33.7 | 44.4 | 35.3 |
| LiVT [63] | ViT-B/16 | 40.8 | 48.1 | 40.6 | 27.5 |
| Fine-tuning CLIP | | | | | |
| Linear Prob (LA) | ViT-B/16 | 48.8 | 48.8 | 49.7 | 47.1 |
| cRT [25] | ViT-B/16 | 44.4 | 51.0 | 43.1 | 35.4 |
| BALLAD [37] | ViT-B/16 | 49.5 | 49.3 | 50.2 | 48.4 |
| Decoder [58] | ViT-B/16 | 46.8 | - | - | - |
| LPT [13] | ViT-B/16 | 50.1 | 49.3 | 52.3 | 46.9 |
| Full-FT (LA) | ViT-B/16 | 46.6 | 49.9 | 46.3 | 41.4 |
| cRT [25] | ViT-B/16 | 44.4 | 51.0 | 43.1 | 35.4 |
| LIFT [50] | ViT-B/16 | **51.5** | 51.3 | 52.2 | **50.5** |
| Model Soups [60] | ViT-B/16 | 49.4 | **51.7** | 50.0 | 43.7 |
| LT-Soups (Ours) | ViT-B/16 | **51.7** | 51.2 | **52.8** | 50.3 |

Table 16: Comparison of methods for training on ImageNet-LT.

| Methods | Backbone | Overall | Many | Medium | Few |
|---------|----------|---------|------|--------|-----|
| Training from scratch | | | | | |
| cRT [25] | ResNet-50 | 47.3 | 58.8 | 44.0 | 26.1 |
| LWS [25] | ResNet-50 | 47.7 | 57.1 | 45.2 | 29.3 |
| MiSLAS [72] | ResNet-50 | 52.7 | 62.9 | 50.7 | 31.0 |
| LA [40] | ResNet-50 | 51.1 | - | - | - |
| DisAlign [67] | ResNet-50 | 52.9 | 61.3 | 52.2 | 31.4 |
| BCL [75] | ResNet-50 | 56.0 | - | - | - |
| PaCo [10] | ResNet-50 | 57.0 | - | - | - |
| NCL [31] | ResNet-50 | 57.4 | - | - | - |
| LiVT [63] | ViT-B/16 | 60.9 | 73.6 | 56.4 | 41.0 |
| Fine-tuning CLIP | | | | | |
| Linear Prob (LA) | ViT-B/16 | 74.2 | 77.8 | 73.3 | 67.4 |
| BALLAD [37] | ViT-B/16 | 75.7 | 79.1 | 74.5 | 69.8 |
| Decoder [58] | ViT-B/16 | 73.2 | - | - | - |
| Full-FT (LA) | ViT-B/16 | 73.9 | 79.8 | 71.9 | 63.9 |
| cRT [25] | ViT-B/16 | 72.6 | 81.1 | 70.6 | 56.1 |
| LIFT [50] | ViT-B/16 | 77.0 | 80.2 | **76.1** | **71.5** |
| Model Soups [60] | ViT-B/16 | 76.0 | **81.5** | 74.5 | 65.5 |
| LT-Soups (Ours) | ViT-B/16 | **77.4** | 81.2 | **76.1** | 70.7 |

Table 17: Comparison of methods for training on NIH-CXR-LT.

| Methods | Backbone | Overall | Many | Medium | Few |
|---------|----------|---------|------|--------|-----|
| Training from ImageNet-1K pre-trained backbone | | | | | |
| cRT [25] | ResNet-50 | 38.0 | 43.3 | 37.4 | 30.0 |
| LWS [25] | ResNet-50 | 28.0 | 45.7 | 23.0 | 08.3 |
| CB LDAM-DRW [5] | ResNet-50 | 37.7 | **47.6** | 35.6 | 25.0 |
| CB Softmax [11] | ResNet-50 | 33.3 | 29.5 | **41.5** | 21.7 |
| Fine-tuning CLIP | | | | | |
| Linear Prob (LA) | ViT-B/16 | 17.5 | 13.3 | 21.1 | 16.7 |
| BALLAD [37] | ViT-B/16 | 34.5 | 36.7 | 38.9 | 20.8 |
| Full-FT (LA) | ViT-B/16 | 38.0 | 43.8 | **41.5** | 20.0 |
| cRT [25] | ViT-B/16 | 37.7 | 42.9 | 39.3 | 25.0 |
| LIFT [50] | ViT-B/16 | 38.5 | 43.3 | 40.4 | 25.5 |
| Model Soups [60] | ViT-B/16 | 38.0 | 45.6 | 40.2 | 20.0 |
| LT-Soups (Ours) | ViT-B/16 | **39.3** | 42.4 | 40.7 | **30.8** |

Table 18: Comparison of methods for training on iNaturalist 2018.

| Methods | Backbone | Overall | Many | Medium | Few |
|---|---|---|---|---|---|
| Training from scratch | | | | | |
| cRT [25] | ResNet-50 | 65.2 | 69.0 | 66.0 | 63.2 |
| LWS [25] | ResNet-50 | 65.9 | 65.0 | 66.3 | 65.5 |
| MiSLAS [72] | ResNet-50 | 71.6 | 73.2 | 72.4 | 70.4 |
| DiVE [18] | ResNet-50 | 69.1 | 70.6 | 70.0 | 67.7 |
| DisAlign [67] | ResNet-50 | 69.5 | 69.1 | 69.9 | 69.4 |
| ALA [71] | ResNet-50 | 69.6 | 69.5 | 70.2 | 69.0 |
| RIDE [57] | ResNet-50 | 71.5 | 72.4 | 73.1 | 70.4 |
| RIDE+CR [38] | ResNet-50 | 73.5 | 74.0 | 74.3 | 73.1 |
| RIDE+OTmix [16] | ResNet-50 | 73.7 | 74.1 | 75.2 | 72.8 |
| BCL [75] | ResNet-50 | 71.8 | - | - | - |
| PaCo [10] | ResNet-50 | 73.2 | 70.4 | 72.8 | 75.8 |
| NCL [31] | ResNet-50 | 74.2 | 72.0 | 74.9 | 73.8 |
| GML [52] | ResNet-50 | 74.5 | - | - | - |
| LiVT [63] | ViT-B/16 | 76.1 | **78.9** | 76.5 | 74.8 |
| Fine-tuning CLIP | | | | | |
| Linear Prob (LA) | ViT-B/16 | 60.4 | 48.9 | 60.0 | 63.9 |
| Decoder [58] | ViT-B/16 | 59.2 | - | - | - |
| LPT [13] | ViT-B/16 | 76.1 | - | - | 79.3 |
| Full-FT (LA) | ViT-B/16 | 76.1 | 75.7 | 76.9 | 75.3 |
| LIFT [50] | ViT-B/16 | **79.1** | 72.4 | **79.0** | **81.1** |
| Model Soups [60] | ViT-B/16 | 76.4 | 77.1 | 76.8 | 75.6 |
| LT-Soups (Ours) | ViT-B/16 | 78.2 | 76.7 | 78.5 | 78.2 |

