# OpenReview forum: "LT-Soups: Bridging Head and Tail Classes via Subsampled Model Soups"
_NeurIPS.cc/2025/Conference — NeurIPS 2025 poster_

### Official Review · Reviewer_g8Rp · 2025-07-01

**Clarity:** 3
**Significance:** 3
**Originality:** 3
**Rating:** 4
**Confidence:** 4

**Summary:**

The paper introduces LT-Soups, a two-stage model soups framework designed to address long-tailed (LT) class imbalance in machine learning. It focuses on the problem where head classes dominate, and tail classes are severely underrepresented, a scenario common in real-world datasets. Previous work on parameter-efficient fine-tuning (PEFT) methods like LoRA and AdaptFormer show improved tail-class performance but often at the cost of head-class accuracy. The authors propose LT-Soups, which averts this trade-off by combining model soups and tailored fine-tuning techniques. The framework works in two stages:

**Questions:**

Real-World Generalization: How does LT-Soups perform when tested on highly skewed datasets or under conditions with noisy labels or missing data? Would the approach still provide strong results?

Statistical Robustness: The paper lacks statistical significance measures such as error bars. Could you provide additional experiments or statistical tests to support the stability of your findings across multiple runs or data splits?

Alternative Subsampling Strategies: Have you considered experimenting with more advanced or adaptive subsampling techniques, such as those based on active learning or dynamic class balancing during training? Could they offer better performance than the current progressive subsampling strategy?

**Ethical Concerns:**

["NO or VERY MINOR ethics concerns only"]

**Final Justification:**

The authors’ response addressed my concerns, and I will maintain my positive rating in the final evaluation.

**Limitations:**

yes

**Quality:**

3

**Strengths And Weaknesses:**

Strengths:
Originality: The introduction of LT-Soups as a two-stage framework that balances performance across both tail and head classes is novel. It builds on the understanding that class imbalance is a complex, multi-dimensional problem and uses a dual-axis framework for imbalance characterization.
Robustness: The method shows significant improvement across both synthetic and real-world long-tailed datasets, including scenarios with varying imbalance ratios and head-tail class distributions.
Practical Application: The proposed method shows better adaptability and performance stability across diverse datasets, making it applicable to real-world tasks, especially those with imbalanced data distributions.

Weaknesses:
Complexity: While the approach is robust, it involves two stages of fine-tuning, which could increase computational costs and complexity compared to simpler methods like PEFT or traditional model soups.
Limited Statistical Analysis: The paper lacks error bars or statistical significance tests in the experimental results, which could make it difficult to assess the variability and robustness of the reported results fully.
Assumptions: The reliance on the head-tail ratio and imbalance ratio, while insightful, might oversimplify the complexity of real-world class distributions. A more flexible or generalized model could be explored to account for other potential distributional shifts.

---

> ### Author Rebuttal · Authors · 2025-07-30
>
> We thank the reviewer for their positive feedback on the originality, robustness, and practical applicability of our method. Below, we address their concerns in detail.
>
> ---
> > ### **Statistical Robustness: The paper lacks statistical significance measures such as error bars. Could you provide additional experiments or statistical tests to support the stability of your findings across multiple runs or data splits?**
> ---
> Please note that we followed the experimental protocol established in prior benchmarks [1,2]. While it is not feasible to run statistical tests across all datasets within the rebuttal timeline, we report results on TinyImageNet-LT using three different random seeds in the table below. As shown, the performance variance is negligible, supporting the robustness of our method. If the reviewers believe it would add value, we would be happy to include statistical tests on the remaining datasets in the final version of the paper.
>
> | **Method**  | **All**     | **Head**    | **Tail**    |
> | ----------- | ----------- | ----------- | ----------- |
> | PEFT        | 77.0 ± 0.31 | 82.7 ± 0.44 | 73.8 ± 0.38 |
> | Model Soups | 77.5 ± 0.04 | 86.0 ± 0.04 | 72.9 ± 0.03 |
> | LT-Soups    | 78.5 ± 0.16 | 85.4 ± 0.4  | 74.6 ± 0.15 |
>
> *[1] Liu et al. CVPR 2019.*
> *[2] Shi et al. ICML 2024.*
>
> ---
> > ### **Real-World Generalization: How does LT-Soups perform when tested on highly skewed datasets or under conditions with noisy labels or missing data? Would the approach still provide strong results?**
> ---
> Please note that two of the benchmark datasets—iNaturalist2018 and NIH-CXR-LT—are real-world datasets with naturally long-tailed distributions. They are characterized by markedly different values of imbalance ratio ($\rho$) and head-to-tail ratio ($\eta$): $\rho=500$, $\eta=0.11$ for iNaturalist2018 and $\rho = 6491$, $\eta = 5.66$ for NIH-CXR-LT. While incorporating additional distribution shifts, such as noisy labels or missing data, is a promising direction for future work, we follow prior work in the long-tailed learning literature [1,2] and focus solely on the imbalance aspect.
>
> *[1] Liu et al. CVPR 2019.*
> *[2] Shi et al. ICML 2024.*
>
> ---
> > ### **Alternative Subsampling Strategies: Have you considered experimenting with more advanced or adaptive subsampling techniques, such as those based on active learning or dynamic class balancing during training? Could they offer better performance than the current progressive subsampling strategy?**
> ---
> Thank you for the suggestion. We agree that more advanced subsampling strategies, such as selecting samples based on information content (e.g., based on entropy), could potentially improve performance and represent a promising direction for future work. However, in this study, we prioritize simplicity to keep our core message focused and interpretable. Notably, even our basic subsampling strategy already outperforms existing baselines, demonstrating the effectiveness of our overall approach.

---

### Official Review · Reviewer_tBqZ · 2025-07-03

**Clarity:** 2
**Significance:** 2
**Originality:** 2
**Rating:** 4
**Confidence:** 3

**Summary:**

They authors propose LT-Soups, a two-stage procedure, where:

1. Progressive subsampling + model-averaging. Multiple CLIP copies are fully fine-tuned (in parallel) on subsets whose imbalance ratios grow geometrically, then exponentially averaged to couple "tail-friendly" and "head-friendly" specialists

2. Classifier re-tuning. Only the final linear head is re-trained on the full data with logit adjustment to regain head-class sharpness

**Questions:**

1. i'd like to see some small ablations if the authors can provide these :)
- Performing PEFT, followed by performing the same final layer only tuning with logit adjustment that is performed in LT-Soups -- would this help in closing gap in head accuracy for PEFT ?
- Performing Model Soups as done in Wortsman et al. [2022], followed by performing the same final layer only tuning

Above two ablations would tell if its the new model soup strategy which is driving the gains or the final logit adjustment.


2. perhaps one more thing to test would be:
- take a subsampled dataset that has the least imbalance, perform vanilla model soups (Wortsman et al. [2022]) on it or PEFT, followed by final layer only tuning -- this ablation would tell us that does one need multiple models at different levels of imbalance in the dataset, or does the least imbalanced dataset is enough

let me know if these make sense.

**Ethical Concerns:**

["NO or VERY MINOR ethics concerns only"]

**Final Justification:**

i'd say the paper is borderline accept.

Reasons for not higher score:
- the novely is a little limited, in the sense, the method seems like a mix-and-match of existing techniques -- this is not a bad thing per say. But mix-and-match in a very specific way adds to the complexity of the method. Overall method is still simple, but its not "aesthetic".

Reasons for not lower score:
- authors provide extensive experiments
- authors did perform the clarifying experiments during rebuttal which helped in understanding
- authors released their code during submission

**Limitations:**

yes

**Quality:**

3

**Strengths And Weaknesses:**

Strengths:
- it's a very simple approach
- it's efficient since it can be parallelized
- consistently outperforms PEFT and classical model-soup baselines on overall, head and tail accuracies
- i really admire that the authors released code in the supplementary


Weakness:
- although the method works well, it's currently hard for me to pinpoint exactly where are the gains coming from -- look at questions section, some ablations will help disentangle the multiple components of the method

---

> ### Author Rebuttal · Authors · 2025-07-30
>
> We thank the reviewer for their insightful feedback and for recognizing the simplicity and effectiveness of our work. Our responses are below:
>
> ---
> > ### **Q1. Performing PEFT and Model soups, followed by performing the same final layer only tuning with logit adjustment that is performed in LT-Soups -- would this help in closing gap in head accuracy for PEFT/Model soups ?**
> ---
> We found that additional final-layer tuning with logit adjustment on PEFT and Model Soups has little to no effect. The table below summarizes the results on TinyImageNet-LT. We hypothesize that, unlike these baselines, LT-Soups does not fully exploit the entire training set, due to the downweighting effect introduced by weight averaging. Consequently, fine-tuning the final layer helps LT-Soups recover head-class sharpness and improves overall performance.
>
> *Table 1: Comparison of baselines with and without Classifier Re-training*
> | **Method**                           | **All** | **Head** | **Tail** |
> | ------------------------------------ | ------: | -------: | -------: |
> | PEFT                                 |    77.1 |     83.0 |     73.9 |
> | PEFT + Classifier re-training        |    77.0 |     83.0 |     73.8 |
> | Model Soups                          |    77.6 |     85.9 |     73.0 |
> | Model Soups + Classifier re-training |    77.6 |     85.5 |     73.4 |
> | LT-Soups Stage 1 (without Classifier re-training)  |    78.1 |     84.9 |     74.5 |
> | LT-Soups                             |    78.6 |     85.0 |     75.2 |
>
> ---
> > ### **Q2. Take a subsampled dataset that has the least imbalance, perform vanilla model soups on it or PEFT, followed by final layer only tuning -- this ablation would tell us that does one needs multiple models at different levels of imbalance in the dataset, or does the least imbalanced dataset is enough**
> ---
> Table 3 in the main paper has already addressed this question in the context of Model Soups. Per your request, we extend the same experiment to the PEFT baseline, with full results shown in the table below.
>
> Specifically, we compare LT-Soups with Model Soups and PEFT baselines that follow the same two-stage framework as LT-Soups. The only difference lies in the first stage: PEFT trains a single model on a subset with a fixed imbalance ratio, while Model Soups averages 16 models, all trained on subsets sharing the same imbalance ratio.
>
> Results show that performance varies significantly depending on the imbalance ratio used. For instance, Model Soups with $\rho = 8$ yields the highest tail accuracy (75.0), while using the full dataset ($\rho = 100$) results in the highest head accuracy (85.9). A similar trade-off is observed with PEFT. However, under any single imbalance setting, both baselines fall short of LT-Soups, which achieves performance of 78.6 (overall), 85.0 (head), and 75.2 (tail).
>
> Rather than optimizing for a single point on the imbalance spectrum, LT-Soups averages across subsets with varying imbalance levels. This enables it to integrate the strengths of both low- and high-$\rho$ models, resulting in a more balanced head–tail trade-off overall.
>
>
> | Imbalance ratio ($\rho$) | 1 | 2 | 4 | 8 | 16 | 32 | 64 | 100 |
> | ------ | - | - | - | - | -- | -- | -- | --- |
> **Model Soups**
> | All | 71.7 | 75.9 | 76.0 | 77.2 | 77.2 | 77.3 | 77.9 | 77.6 |
> | Head | 74.6 | 78.6 | 78.7 | 81.0 | 82.8 | 84.7 | 85.5 | 85.5 |
> | Tail | 70.1 | 74.4 | 74.6 | 75.0 | 74.1 | 73.3 | 73.7 | 73.4
> **PEFT**
> | All | 73.9 | 74.3 | 74.8 | 76.2 | 76.4 | 77.1 | 77.0 | 77.0 |
> | Head | 75.9 | 75.8 | 77.0 | 78.2 | 80.4 | 81.8 | 81.9 | 83.0 |
> | Tail | 72.8 | 73.5 | 73.6 | 75.1 | 74.2 | 74.5 | 74.4 | 73.8 |

---

> > ### Comment · Reviewer_tBqZ · 2025-08-04
> >
> > Thanks for providing these!
> >
> > I’d say add a discussion in the paper about this — it will help clarify things to the reader about what is important in making LT soups work.
> >
> > I will maintain my positive score, thanks :)

---

> > > ### Author Response · Authors · 2025-08-04
> > >
> > > Thank you for the suggestion; we will include it in the final version. Thanks for appreciating our work.

---

### Official Review · Reviewer_9QKC · 2025-07-03

**Clarity:** 3
**Significance:** 2
**Originality:** 2
**Rating:** 4
**Confidence:** 4

**Summary:**

This paper addresses the trade-off between head- and tail-class performance when fine-tuning large pre-trained models on long-tailed (LT) datasets. The authors begin by introducing a new metric, the head-tail ratio (η), which they use alongside the standard imbalance ratio (ρ) to more comprehensively characterize the structure of long-tailed distributions. Through experiments, they show that Parameter-Efficient Fine-Tuning (PEFT) methods excel in tail-heavy scenarios, while full fine-tuning is more advantageous when head classes are dominant.

To resolve this issue, the authors propose LT-Soups, a two-stage model merging framework. In the first stage, instead of training models on the entire imbalanced dataset, LT-Soups creates a series of data subsets with progressively increasing imbalance ratios. It fine-tunes a full model on each subset and then merges the weights of these "specialist" models via recursive weighted averaging to form a unified feature extractor. In the second stage, to recover head-class information potentially lost during subsampling, the framework freezes the merged backbone and retrains only the classifier layer on the full training set. Experimental results across several synthetic and real-world long-tailed datasets demonstrate that LT-Soups achieves a superior performance balance compared to PEFT and traditional model soups across diverse long-tailed regimes.

**Questions:**

None

**Ethical Concerns:**

["NO or VERY MINOR ethics concerns only"]

**Final Justification:**

I thank the authors for their detailed and timely response. After reviewing their feedback, I believe that most of my concerns have been adequately addressed. I have therefore decided to raise my original rating, leaning toward acceptance.

**Limitations:**

yes

**Quality:**

2

**Strengths And Weaknesses:**

**Strengths**
1. The paper's analysis of the head-tail performance trade-off is insightful. The introduction of the head-tail ratio (η) as a new analytical dimension provides a more fine-grained framework for understanding the complexities of long-tailed recognition.

2. The paper's experimental section is robust. The authors validate their method on a wide range of standard benchmarks and use well-designed ablation studies to demonstrate the contribution of each component. The visualizations effectively support the core claim that different methods are optimal for different imbalance structures.

**Weaknesses**
1. While LT-Soups is an effective framework, its core ideas are more of an intelligent combination of existing techniques—Model Soups, data subsampling, and two-stage training—rather than a fundamentally new concept. The innovation lies in the specific recipe, which, while successful, may not meet the high bar for conceptual novelty at a top-tier venue.

2. The framework requires training N×M models in the first stage. Although parallelizable and trained on smaller subsets, the total computational budget is significantly higher than for single-run methods like PEFT or Full-FT. A direct comparison of total GPU hours is missing, making it difficult to assess the practical cost-benefit trade-off.

3. The recursive, ordered merging strategy is shown empirically to be superior to alternatives. However, the paper lacks a deep, intuitive explanation for why this curriculum-like fusion (from balanced to imbalanced) is the optimal approach. It remains unclear if this is a robust principle or an empirical finding specific to this setup.

4. Lack of discussion on recent literature. For ensemble long-tail recognition: Self-supervised aggregation of diverse experts for test-agnostic long-tailed recognition, Mdcs: More diverse experts with consistency self-distillation for long-tailed recognition. For recent literature: LTRL: Boosting Long-tail Recognition via Reflective Learning, DiffuLT: Diffusion for Long-tail Recognition Without External Knowledge, Harnessing Hierarchical Label Distribution Variations in Test Agnostic Long-tail Recognition.

---

> ### Author Rebuttal · Authors · 2025-07-30
>
> We thank the reviewer for their thoughtful feedback and encouraging comments on our analysis, rigorous methodology, and well-designed ablation studies. We address their questions below:
>
> ---
> > ### **W1. While LT-Soups is an effective framework, its core ideas are more of an intelligent combination of existing techniques.**
> ---
> We emphasize that our contribution goes beyond simply proposing a final model. Specifically:
>
> 1. ***A new lens for analyzing long-tailed learning:*** We introduce the head-to-tail ratio ($\eta$) as a complementary axis to the standard imbalance ratio ($\rho$), offering a more comprehensive view of class imbalance. We argue that effective long-tailed methods must perform well across the full ($\rho$, $\eta$) spectrum—*a perspective that enhances understanding of real-world imbalance*. Through this lens, we examine state-of-the-art methods such as a very strong competitor, LIFT [1], which advocates parameter-efficient fine-tuning (PEFT) for long-tailed tasks, and Model Soups [2], which uniformly average fully fine-tuned models on the full imbalanced dataset. This analysis reveals performance trade-offs that are often have overlooked in conventional evaluations.
> 2. ***LT-Soups (a principled adaptation of Model Soups):*** Building on these insights, we propose LT-Soups, which adapts the strengths of Model Soups to long-tailed settings. Model Soups boosts head-class performance but significantly degrades tail-class accuracy due to overexposure to head-heavy distributions. PEFT, on the other hand, limits parameter updates and struggles to adapt to head classes. LT-Soups addresses both issues by fully fine-tuning models on subsets with progressively increasing imbalance and combining them via recursive weight averaging. This balances head-class adaptation with tail-class generalization
> 3. ***Comprehensive evaluation:*** Across six benchmark datasets, LT-Soups outperforms PEFT in overall accuracy (by 1.3 to 2.3 percentage points) on five of them, while achieving comparable tail-class performance. It also consistently surpasses Model Soups in overall accuracy (by 1.3 to 2.3 points) and yields substantial gains in tail-class accuracy (by 2.6 to 10.9 points), with only a modest drop in head-class performance (up to 3.2 points).
>
> *[1] Shi et al. ICML 2025.*
> *[2] Wortsman et al. ICML 2022.*
>
> ---
> > ### **W2. A direct comparison of total GPU hours is missing, making it difficult to assess the practical cost-benefit trade-off.**
> ---
> The full computational analysis, including total wall-clock time for the parallel version of LT-Soups, is already provided in Appendix B. In response to the reviewers’ request, we also report the total GPU hours, assuming access to only a single GPU with modest memory.
>
> To recap, LT-Soups consists of two stages: Stage 1 trains a series of specialist models on subsampled datasets with varying imbalance ratios. Stage 2 involves retraining only a linear classifier on the full dataset, which is computationally lightweight. Both stages use the same number of epochs.
>
> Tables 1 and 2 report the Stage 1 fine-tuning time per subsample (each $\rho$ corresponds to one subsample) and the total GPU hours when LT-Soups is trained on a *single GPU* for the ImageNet-LT and NIH-CXR-LT datasets. As we can see, the computational cost of LT-Soups depends primarily on the degree of imbalance in the dataset. For example, in ImageNet-LT (which comes with \rho=256), following our subsampling strategy, the total absolute GPU time is ~8 hours. LT-Soups is costly in this case because with our largest \rho=64, it includes 80% of the entire training data. In contrast, in NIH-CXR-LT ($\rho = 6401$), the largest subset ($\rho = 256$) includes only 24% of the data, ***resulting in substantial savings, up to 18x lower GPU hours compared to Model Soups of the same size.***
>
> *Table 1: Computational cost per ImageNet-LT subset for LT-Soups with $M=2$.*
> | $\rho$        | 1       | 2       | 4       | 8       | 16      | 32      | 64      | **Total GPU hours** |
> | ------------- | ------- | ------- | ------- | ------- | ------- | ------- | ------- | ------- |
> | **GPU Hours** | 0:12:19 | 0:18:44 | 0:32:00 | 0:54:00 | 1:26:32 | 2:02:00 | 2:31:09 | 7:56:50 |
>
>
> *Table 2: Computational cost per CXR-LT subset for LT-Soups with $M=2$.*
> | $\rho$        | 1       | 2       | 4       | 8       | 16      | 32      | 64      | 128     | 256     | **Total GPU hours** |
> | ------------- | ------- | ------- | ------- | ------- | ------- | ------- | ------- | ------- | ------- | ------- |
> | **GPU Hours** | 0:01:01 | 0:01:05 | 0:01:21 | 0:02:38 | 0:04:06 | 0:06:16 | 0:10:26 | 0:16:46 | 0:25:04 | 1:08:12 |
>
> We conclude by referencing Table 9 in the Appendix, which compares the total computational cost of LT-Soups and Model Soups *(both with parallelization)* against other baselines on ImageNet-LT and NIH-CXR-LT. *Notably, although LT-Soups is a full-rank method, it converges within 10 epochs on CXR-LT, while LoRA—despite its parameter efficiency—requires 50 epochs, resulting in longer wall-clock time and higher total GPU hours*. Thus, in response to the reviewer’s concern, the claim that “the total computational budget is significantly higher than single-run methods” does not always hold; LT-Soups often reduces computational cost through subsampling, with the budget depending on the choice of $N$ (Tab. 8 Appendix).
>
> | **Method**       | **Wall-clock time (H\:M\:S)** | **Epochs** |
> | ---------------- | ----------------------------- | ---------- |
> | **ImageNet-LT**  |                               |            |
> | Full-FT          | 1:37:56                       | 10         |
> | Model Soups      | 1:37:56                       | 10         |
> | LoRA (rank=64)   | 1:25:33                       | 10         |
> | LT-Soups         | 1:45:38                       | 10         |
> | **CXR-LT**       |                               |            |
> | Full-FT          | 0:53:43                       | 10         |
> | Model Soups      | 0:53:43                       | 10         |
> | LoRA (rank=64)   | 2:14:32                       | 50         |
> | LT-Soups         | 0:32:17                       | 10         |
>
> ---
> > ### **W3. The paper lacks a deep, intuitive explanation for why this curriculum-like fusion (from balanced to imbalanced) is the optimal approach.**
> ---
> We believe our recursive weight averaging (WA) is a principled approach, as it enables an effective trade-off between stronger adaptation—potentially biased toward head classes—and more balanced generalization. It can be interpreted as an exponential moving average (EMA) over fine-tuned models sorted by increasing imbalance severity, with a tunable parameter that adjusts the influence of more balanced (but smaller) versus less balanced (but larger) subsets. In contrast, uniform WA applies a simple arithmetic mean, giving equal weight to all models regardless of their imbalance level.
>
> In all of our experiments in the paper, we use only two values for $\lambda$: 0.3 and 0.7, corresponding to high and low adaptation needs, respectively. Intuitively, when the target dataset is close to the pre-training weights, the value of the $\lambda$ becomes less important as even small datasets are enough for adaptations. However, when the shift becomes larger, subsets with more data (albeit biased towards head classes) become crucial.
>
> The table below confirms our hypotheses. In particular, we compare recursive WA and uniform WA across two datasets with different similarities compared to CLIP-pretrained weights (according to the zero-shot performance). On ImageNet-LT, which is already well-aligned with CLIP-pretrained features, there is little to no difference between the two averaging schemes. However, for datasets that require significant adaptation [1], such as iNaturalist2018, recursive WA yields clear benefits by leveraging information from more data-rich subsets.
>
>
> *Table 1: Performance comparison under uniform and recursive WA across three datasets.*
> | Method      | **TinyImageNet-LT** |      |      |      | **iNat2018** |      |      |      |
> | ----------- | ------------------- | ---- | ---- | ---- | ------------ | ---- | ---- | ---- |
> |             | All                 | Many | Med. | Few  | All          | Many | Med. | Few  |
> | **Uniform** | 78.5                | 83.4 | 78.4 | 72.9 | 74.7         | 67.4 | 75.8 | 75.3 |
> | **Ours**    | 78.6                | 85.0 | 78.3 | 71.5 | 78.2         | 76.7 | 78.5 | 78.2 |
>
> ---
> > ### **W4. Lack of discussion on recent literature.**
> ---
> Thank you for the suggestions. Given that our focus was on using pre-trained CLIP models for LT recognition, we have restricted the comparison to the recent literature that uses CLIP for LT recognition. In response to the reviewer’s request, we have expanded the related work section and now provide a brief comparison of LT-Soups with the methods mentioned.
>
> SADE [1], Mdcs [2], and DirMixE [5] are Mixture-of-Experts (MoE) methods that aggregate diverse experts trained with different logit adjustment (LA), targeting distributions like uniform, long-tail, and inverse long-tail. Unlike these methods, which require deploying all experts at inference, LT-Soups uses weight averaging to collapse models into a single, inference-efficient solution. Moreover, as noted in [6], fully fine-tuning foundation models like CLIP with LA loss alone is insufficient, as it can lead to inconsistent class-conditional distributions, particularly for tail classes.
>
> Reflective Learning [3] promotes consistency across training iterations by minimizing KL divergence between predictions and soft labels induced from feature similarity. In this light, the EMA mechanism in LT-Soups can be viewed as a lightweight form of Reflective Learning or self-distillation [7].
>
> *[1] Zhang et al., NeurIPS 2022.*
> *[2] Zhao et al., ICCV 2023.*
> *[3] Zhao et al., ECCV 2024.*
> *[5] Yang et al., ICML 2024.*
> *[6] Shi et al., ICML 2024.*
> *[7] Allen-Zhu et al., ICLR 2023.*

---

> ### Comment · Reviewer_9QKC · 2025-08-05
>
> I thank the authors for their detailed and timely response. After reviewing their feedback, I believe that most of my concerns have been adequately addressed. I have therefore decided to raise my original rating, leaning toward acceptance.

---

### Official Review · Reviewer_Q5RH · 2025-07-07

**Clarity:** 2
**Significance:** 2
**Originality:** 2
**Rating:** 3
**Confidence:** 1

**Summary:**

As I notified the AC in June 2nd, I deeply apologize but due to an unexpected change in my professional schedule, I will not be able to perform my reviews this year.

**Questions:**

As I notified the AC in June 2nd, I deeply apologize but due to an unexpected change in my professional schedule, I will not be able to perform my reviews this year.

**Ethical Concerns:**

["NO or VERY MINOR ethics concerns only"]

**Final Justification:**

Please ignore my rating. I still don't understand why I have to go through this even though i notified early on that i would not able to perform my reviews this year.

**Limitations:**

As I notified the AC in June 2nd, I deeply apologize but due to an unexpected change in my professional schedule, I will not be able to perform my reviews this year.

**Paper Formatting Concerns:**

As I notified the AC in June 2nd, I deeply apologize but due to an unexpected change in my professional schedule, I will not be able to perform my reviews this year.

**Quality:**

2

**Strengths And Weaknesses:**

As I notified the AC in June 2nd, I deeply apologize but due to an unexpected change in my professional schedule, I will not be able to perform my reviews this year.

---

### Official Review · Reviewer_snjd · 2025-07-09

**Clarity:** 2
**Significance:** 3
**Originality:** 2
**Rating:** 4
**Confidence:** 3

**Summary:**

In this paper, the authors conduct experiments to show that previous methods for the long-tail problem cannot achieve better performance both on head and tail classes. Therefore, the authors propose to adapt the model soup for the problem by training different models on sub-datasets with different class imbalance ratios and averaging their weights to obtain better representation.

**Questions:**

Please refer to the weakness part.

**Ethical Concerns:**

["NO or VERY MINOR ethics concerns only"]

**Final Justification:**

The authors' response solve most of my problems. Considering the simplicity but effectiveness of the proposed method, I recommend acceptance.

**Limitations:**

Please refer to the weakness part.

**Paper Formatting Concerns:**

There is no paper formatting problem.

**Quality:**

2

**Strengths And Weaknesses:**

Strength:
1. The authors identify the problem of head-tail imbalance ratio which is neglected by previous works.
2. The motivation of the paper is stated clearly by conducting preliminary experiments.

Weakness:
1. The writing is not proper enough. For example, throughout the paper, how the Logit Adjustment is performed is not mathematically formulated. Although it has been proposed in previous works, rephasing it in the current paper helps others understand it better. The caption of Figure 5 is neglected as well. The momentum coefficient $\mu$ in line 210 is not defined either. If it is the same to $\lambda$ above, then the last weight of the averaged model is almost the same as the first model so that the proposed methods is meaningless.
2. It seems that the proposed method cannot solve the targeted problem. From the experiments, the proposed methods can improve the performance of model soup on tail classes, but the performance on head classes still degrades which is the problem that the authors stated to solve.
3. The experimental setting is not comprehensive enough. For example, one of the setting of imbalance ratio or head-tail ratio in Table 1 and 2 should be kept when the other one is changing. Moreover, the results illustrated in Figure 2 is not convincing enough because they are averaged on head-tail ratio which will not be able to show the detailed results when the head-tail ratio is high or low.
4. Can you show how the performance changes when the total number of models changes?

---

> ### Author Rebuttal · Authors · 2025-07-30
>
> We thank the reviewer for their feedback and address their concerns in detail below:
>
> ---
> > ### **W2. It seems that the proposed method cannot solve the targeted problem. From the experiments, the proposed methods can improve the performance of model soup on tail classes, but the performance on head classes still degrades which is the problem that the authors stated to solve.**
> ---
> It is well established in the class imbalance literature [1,2,3,4] that, given an imbalanced dataset, the goal is to maximize balanced accuracy (see L122–124) on a uniformly distributed test set. This, in turn, requires achieving a well-balanced performance across both head and tail classes.
>
> While Model Soups favours head-class performance by fully fine-tuning on the original imbalanced distribution, this improvement comes at the cost of reduced accuracy on tail classes. On the other hand, PEFT limits weight updates to a small subset of parameters, which negatively impacts head-class performance. In contrast, LT-Soups fine-tunes and averages models trained on distributions with progressively increasing imbalance ratios. This strategy not only allows adaptation to head classes but also preserves the balanced representations needed for tail-class generalization. Across all six benchmarked datasets, LT-Soups consistently outperforms Model Soups in overall accuracy (by 1.3 to 2.3 percentage points) and shows substantial improvements in tail-class accuracy (by 2.6 to 10.9 points), with only a modest decrease in head-class performance (up to 3.2 points).
>
> *[1] Cao et al. Neurips 2019.*
> *[2] Ren et al. Neurips 2020.*
> *[3] Menon et al. ICLR 2020.*
> *[4] Wang et al. ICLR 2020.*
>
> ---
> > ### **W3. The experimental setting is not comprehensive enough. For example, one of the setting of imbalance ratio or head-tail ratio in Table 1 and 2 should be kept when the other one is changing.**
> ---
>
> Please note that we followed the experimental protocol established in prior benchmarks [1,2]. Particularly, we conduct a comprehensive analysis across six datasets: four synthetic long-tailed distributions, CIFAR100-LT, ImageNet-LT, TinyImageNet-LT, and Places-LT, and two real-world long-tailed distributions, NIH-CXR-LT and iNaturalist2018. Each dataset features a distinct combination of imbalance ratio and head-to-tail ratio, providing a diverse representation of imbalance scenarios.
>
> Furthermore, in Section 3.2 of the paper, we systematically vary the imbalance ratio ($\rho$) across three levels and the head-to-tail ratio ($\eta$) across eleven values by synthetically modifying the CIFAR-100 distribution. Each configuration is evaluated across four baselines, including LT-Soups, resulting in a total of 396 training runs. While we expect the trends observed on CIFAR-100 to generalize to other datasets such as TinyImageNet-LT and those in Tables 1 and 2, we are happy to include additional results in the Appendix if the reviewer believes it would strengthen our findings.
>
> [1] Liu et al. CVPR 2019. [2] Shi et al. ICML 2024.
>
> ---
> > ### **W4. Can you show how the performance changes when the total number of models changes?**
> ---
> Increasing the total number of models consistently improves overall performance, as it has already been presented in Figure 5b and L296 in the main paper and Table 7 in the Appendix. For the reviewer’s convenience, we summarize these results in the two tables below.
>
> To recap, the total number of models used in LT-Soups is governed by two key hyperparameters: $N$, the number of subsamples with varying imbalance ratios, and $M$, the number of models trained per subsample.
>
> Table 1 explores the effect of increasing the total number of models by varying $N$, from the most balanced case ($N=1$) to the full dataset ($N=8$) while keeping the number of models per subsample fixed at $M=2$. Table 2 examines the impact of increasing $M$ while holding $N=8$ fixed. Overall, we observe consistent improvements in performance as the total number of models increases. To keep the experiments manageable, we fix $M = 2$ in the main benchmarks.
>
>
> *Table 1: Effect of increasing the total models on performance by varying the number of subsamples ($N$)*
> |                  | N = 1 | N = 2 | N = 3 | N = 4 | N = 5 | N = 6 | N = 7 | N = 8 |
> | ---------------- | ----- | ----- | ----- | ----- | ----- | ----- | ----- | ----- |
> | **Total Models** | 2     | 4     | 6     | 8     | 10    | 12    | 14    | 16    |
> | **All**          | 71.7  | 74.0  | 75.9  | 77.9  | 77.6  | 78.0  | 78.1  | 78.6  |
> | **Head**         | 74.6  | 76.6  | 78.5  | 81.2  | 82.0  | 83.0  | 84.5  | 85.0  |
> | **Tail**         | 70.1  | 72.5  | 74.5  | 76.1  | 75.2  | 75.4  | 74.7  | 75.2  |
>
> *Table 2:  Effect of increasing the total models on performance by varying the number of models trained per subsample ($M$)*
> |                  | M = 1 | M = 2 | M = 12 |
> | ---------------- | ----- | ----- | ------ |
> | **Total Models** | 8     | 16    | 96     |
> | **All**          | 78.2  | 78.6  | 78.8   |
> | **Head**         | 84.8  | 85.0  | 85.5   |
> | **Tail**         | 74.6  | 75.2  | 75.5   |
>
> ---
> > ### **W1.1. The results illustrated in Figure 2 is not convincing enough because they are averaged on head-tail ratio which will not be able to show the detailed results when the head-tail ratio is high or low.**
> ---
> We would like to clarify that the analysis requested by the reviewer corresponds to Figure 1 of the main paper. This figure reports the performance of four methods, including LT-Soups, on synthetic CIFAR100 across three levels of imbalance ratio ($\rho$) and eleven head-to-tail ratio ($\eta$) values. Figure 2 is derived from the same data by marginalizing over both $\rho$ and $\eta$ to emphasize a central insight of our work: an effective LT method must perform well across the full spectrum of imbalance conditions.
>
> ---
> > ### **W1.2. Throughout the paper, how the Logit Adjustment is performed is not mathematically formulated. Although it has been proposed in previous works, rephasing it in the current paper helps others understand it better.**
> ---
> We appreciate your suggestion and will revise the description of the Logit Adjustment loss function in Section **3.1** for greater clarity.
>
> ---
> > ### **W1.3. The caption of Figure 5 is neglected as well.**
> ---
> To improve clarity, we split the original **Figure 5** into two separate figures, each with its own caption.
>
> ---
> > ### **W1.4. The momentum coefficient $\mu$ in line 210 is not defined either. If it is the same as $\lambda$ above, then the last weight of the averaged model is almost the same as the first model, so that the proposed method is meaningless.**
> ---
> We use the following formulation for EMA:
>
> $\theta_{ema} = (1-\mu) \cdot \theta_{ema} + \mu \cdot \theta_0,$
>
> where $\theta_0$ is the initialization weights and $\theta_{ema}$ denotes the fine-tuning parameters. We fixed $\mu=0.01$ throughout all the experiments. Therefore, the last weight of the averaged model is clearly not the same as the first model. We will clarify this in the revised manuscript.

---

> ### Comment · Reviewer_snjd · 2025-08-06
>
> Thanks to the authors' responses. Most of my problems have been solved. Sorry for that I overlooked some parts of the paper and had some misunderstanding. I have raised my rating to recommend acceptance.

---

### Comment · Area_Chair_Rdmq · 2025-08-04

Dear Reviewers,

Thank you very much for your time and efforts. As we are approaching the deadline, we kindly ask you to review the rebuttal and share any remaining concerns with the authors for discussion.

Best regards, AC

---

### Note · Authors · 2025-08-13

We thank all reviewers for their constructive feedback. Following their requests:

- We clarified the motivation, implementation details, and experimental setup, addressing earlier misunderstandings.

- We added discussion on the motivation and empirical robustness of the ordered merging strategy, incorporated additional relevant literature, and expanded the computational analysis to include worst-case GPU-hour costs.

- We provided ablations disentangling the contributions of progressive subsampling and final classifier re-tuning, confirming that the performance gains arise from the synergy of these components.

- We repeated experiments on TinyImageNet three times to show the statistical robustness of our method.

We will integrate the above contents into the final version of the paper, as suggested by the reviewers.

Following our rebuttal, reviewer **snjd** raised their rating, stating “*I have raised my rating to recommend acceptance.*”. Similarly, reviewer **9QKC** increased their rating, adding “*I have therefore decided to raise my original rating, leaning toward acceptance.*”. The reviewer **tBqZ** maintained their already positive score. We hope that reviewer **g8Rp**, who initially *provided a positive rating*, was also satisfied, as no follow-up comments were received.

Finally, we note that reviewer **Q5RH** *explicitly stated they did not review the paper, yet assigned a rating of 3*. We believe this score should not be considered in the final decision.

We thank the AC for the time and consideration.

---

### Decision · Program_Chairs · 2025-09-17

**Decision:**

Accept (poster)

**Comment:**

Despite some concerns about conceptual novelty and computational efficiency, the paper makes a meaningful and well-validated contribution to long-tailed recognition with foundation models. The introduction of the head-tail ratio as an analytical tool and the empirical success of LT-Soups across diverse imbalance regimes provide strong reasons for acceptance. The authors’ detailed rebuttal and additional experiments effectively addressed reviewer concerns, and multiple reviewers raised their scores accordingly. The Author should address all reviewers' concerns in their final version.